# Therapeutic Potential of Natural Compounds in Neurodegenerative Diseases: Insights from Clinical Trials

**DOI:** 10.3390/pharmaceutics15010212

**Published:** 2023-01-07

**Authors:** Stéphanie Andrade, Débora Nunes, Meghna Dabur, Maria J. Ramalho, Maria C. Pereira, Joana A. Loureiro

**Affiliations:** 1LEPABE—Laboratory for Process Engineering, Environment, Biotechnology and Energy, Faculty of Engineering, University of Porto, Rua Dr. Roberto Frias, 4200-465 Porto, Portugal; 2ALiCE—Associate Laboratory in Chemical Engineering, Faculty of Engineering, University of Porto, Rua Dr. Roberto Frias, 4200-465 Porto, Portugal

**Keywords:** natural product, dietary supplement, cognitive function, human study, clinical study

## Abstract

Neurodegenerative diseases are caused by the gradual loss of neurons’ function. These neurological illnesses remain incurable, and current medicines only alleviate the symptoms. Given the social and economic burden caused by the rising frequency of neurodegenerative diseases, there is an urgent need for the development of appropriate therapeutics. Natural compounds are gaining popularity as alternatives to synthetic drugs due to their neuroprotective properties and higher biocompatibility. While natural compounds’ therapeutic effects for neurodegenerative disease treatment have been investigated in numerous in vitro and in vivo studies, only few have moved to clinical trials. This article provides the first systematic review of the clinical trials evaluating natural compounds’ safety and efficacy for the treatment of the five most prevalent neurodegenerative disorders: Alzheimer’s disease, Parkinson’s disease, multiple sclerosis, amyotrophic lateral sclerosis, and Huntington’s disease.

## 1. Introduction

Neurodegenerative diseases refer to disorders that predominantly damage the neurons. These incurable and irreversible disorders cause nerve cells to degenerate or die over time, which are necessary for mobility, strength, sensation, and cognition. Some medications are being prescribed to attenuate the symptoms of neurodegenerative diseases [1]. These medicines, in addition to not offering a cure, cause a variety of side effects such as headache, dizziness, and nausea [1].

Over the years, natural compounds have arisen increasing interest as promising alternatives to synthetic drugs due to their presumed safety for human intake since they are widely consumed on a daily basis, facilitating their clinical approval. A variety of natural products from various origins has been proposed for neurodegenerative disease therapy due to their neuroprotective benefits [2]. Although hundreds of natural compounds have been identified for neurodegenerative diseases therapy in preclinical studies, only a small number has progressed to clinical trials to validate their safety and efficacy. 

This article provides the first systematic review of natural compounds whose effectiveness in the treatment of neurodegenerative diseases has been evaluated in clinical trials. To cover a wide range of topics, the five most prevalent neurodegenerative diseases—Alzheimer’s disease (AD), Parkinson’s disease (PD), multiple sclerosis (MS), amyotrophic lateral sclerosis (ALS), or Huntington’s disease (HD)—were included in this review. Section 2 provides a brief description of the aforementioned diseases. Section 3 describes the clinical trials assessing the natural compounds’ efficacy in treating those illnesses. In case there are several clinical trials for the same molecule, the clinical trial in the most advanced phase is described. Lastly, the discussion of the described studies, the main challenges of the field, and concluding remarks are presented in Section 4. Key words used in all databases include the name of each disease (i.e., Alzheimer’s disease, Parkinson’s disease, multiple sclerosis, amyotrophic lateral sclerosis, and Huntington’s disease) combined with “natural compound”, “natural product”, or the name of 60 natural compounds, which are listed in the Appendix A. Plant extracts were excluded from this investigation. The literature search was conducted using ClinicalTrials.gov, PubMed, Science direct, Google Scholar, Scopus, and Web of Science as online databases until October 2022. Only trials reported in English were considered with no date limit.

## 2. Most Prevalent Neurodegenerative Diseases

Neurodegenerative diseases affect both the central nervous system (CNS) and the peripheral nervous system [3]. Neurodegeneration is triggered by multiple factors, including genetic and environmental ones. Neurodegenerative diseases are classified according to the anatomic distribution of neurodegeneration, molecular pathogenesis, and initial clinical symptoms [4]. AD, PD, MS, ALS, and HD are the five most prevalent neurodegenerative disorders.

AD is the most common neurodegenerative disease and the leading cause of dementia in the elderly, accounting for around 70% of dementia cases globally [5]. Aging is the most well-known risk factor for AD, with almost half of the world’s population over 75 diagnosed with this disease [6]. The World Health Organization (WHO) estimates that more than 55 million people suffer from AD globally [7]. AD is characterized by severe loss of cognitive functions and declarative memory, communication issues, and psychiatric symptoms such as apathy, spatial disorientation, general disinterest, depression, agitation, and aggression. AD remains a progressive disease and, so far, is irreversible [8]. The amyloid cascade hypothesis (ACH) has been the most influential theory behind AD pathology. The ACH postulates that the neurodegeneration observed in AD is caused by the abnormal accumulation of amyloid-β (Aβ) plaques in various brain areas [9]. The formation of these diffuse plaques results in neuroinflammation, oxidative stress, and neurofibrillary tangle (NFTs) formation containing hyperphosphorylated tau protein. NFTs accumulate inside the neurons leading to neurotransmitter system deficits, synaptic dysfunction, and neuronal loss in critical areas associated with cognitive functions, such as memory. This cascade of events culminates in cell death leading to dementia [10]. 

PD is the second most common neurodegenerative disorder [11]. The disease affects mainly adults over the age of 66 and is more prevalent in men than women [12]. According to the WHO, more than 10 million people worldwide were diagnosed with PD [13]. PD mainly affects the patients’ muscles and motor function. Resting tremors, akinesia (lack of spontaneous voluntary movement), bradykinesia (slowness of movement), dyskinesia (involuntary movements), and postural instability are the main outward signs of PD [14]. The non-motor symptoms include psychiatric problems, autonomic dysfunctions, cognitive impairment, and sleep disorders [15]. Although the pathophysiology of PD is still not fully understood, it is known that Lewy bodies with aggregated α-synuclein protein [15] and the loss or degeneration of dopamine-producing neurons (one of the brain’s dopamine pathways) are related to this disease [16]. Neuroinflammation, mitochondrial dysfunction, and oxidative stress are also known leading causes of PD [12,14,15]. 

MS is the third most prevalent neurodegenerative disease in the world. It affects approximately 2.8 million people worldwide [17]. The disease occurs twice more often in women, and the diagnosis usually occurs in young people aged between 20 and 50, with the mean age of diagnosis being 32 years [18]. Despite its unknown cause, MS pathology has been suggested to be of autoimmune nature. Autoreactive T cells trigger the local inflammation of the CNS, leading to the demyelination of nerve cells, glial scar formation (gliosis), and axonal loss [19]. MS causes different symptoms, including psychiatric manifestations such as depression and personality disorder [20], and physical manifestations such as fatigue, muscle weakness, and ataxia [21]. With illness progression, more severe manifestations occur in higher prevalence, such as impaired speech and vision, and cognitive dysfunction that leads to deficits in attention, memory, and language, and paralysis [20]. 

ALS, often called Lou Gehrig’s disease, is the fourth most common neurodegenerative disorder and one of the most frequent motor neuron diseases. Nowadays, it affects approximately 290,000–360,000 patients worldwide [22]. Although its origin is still unknown [23], this neurodegenerative disorder is believed to be caused by genetic, age-related, and environmental factors and is more prevalent in the male gender [24]. It is characterized by progressive muscle weakness derived from upper and lower motor neuron loss, causing loss of muscle control. The disease is also characterized by persistent fatigue, uncontrollable periods of laughing or crying, and trouble in projecting the voice [25]. The average survival after diagnosis is between 3 (up to 70% of the patients) to 5 years, which may be a slightly higher when progression is slower [26].

HD is the fifth most common neurodegenerative disease and the most prevalent inherited neurodegenerative illness, with uncontrolled excessive motor movements as well as cognitive and emotional abnormalities [27]. The current global prevalence of HD is 4 per 100,000 persons [28] and affects approximately 300,000 patients yearly (based on the 2022 world’s population data) [29]. The average onset age is 40, with mortality occurring 15–20 years after the disease has been diagnosed [30]. HD results from the mutation in the Huntingtin (Htt) gene. This mutation causes an excessively long polyglutamine, which confers one or more lethal activities on mutant Htt, resulting in neurodegeneration. The typical function of Htt is unknown; however, it may be involved in internal cell signaling, cyclic adenosine monophosphate response element binding protein maintenance, and neuronal toxicity prevention [31]. 

All the available pharmacological therapies for the described neurodegenerative disorders are ineffective in reversing the pathological changes that underlie these diseases and only provide symptomatic relief for patients. Besides, most of them induce undesirable side effects such as headache, constipation, confusion, dizziness, nausea, vomiting, loss of appetite, and an increased frequency of bowel movements [1]. Therefore, it is fundamental to seek new strategies for these diseases.

## 3. Role of Natural Compounds in the Treatment of Neurodegenerative Diseases

Natural compounds have sparked increased interest as promising alternatives to synthetic medications over the years [32]. Due to their neuroprotective effects, a diversity of natural compounds from different origins has been proposed for treating neurodegenerative diseases. They have been demonstrated to target the various aspects of disease pathogenesis rather than just the symptoms [33].

Natural compounds are classified into four groups according to their origin: alkaloids, phenylpropanoids, polyketides, and terpenoids. *Alkaloids* are a class of naturally occurring organic nitrogen-containing compounds [34]. They are found in many plants, animals, bacteria, and fungi and are recognized for their anti-inflammatory, antioxidant, anticancer, analgesic, antimicrobial, and antifungal activities. Caffeine, nicotine, and huperzine A are examples of alkaloids that have been investigated for their therapeutic relevance in treating neurogenerative diseases. *Phenylpropanoids*, also known as cinnamic acids, are a group of natural compounds with an aromatic ring and a three-carbon propene tail synthesized by plants from phenylalanine or tyrosine amino acids. Phenylpropanoids can be classified into five groups: flavonoids, lignins, phenolic acids, stilbenes, and coumarins [35]. In recent years, much interest has been attracted regarding phenylpropanoids such as resveratrol, curcumin, and epigallocatechin gallate (EGCG) for the therapy of neurodegenerative diseases, due to their antioxidant and anti-inflammatory properties [36]. On the contrary, there is little evidence of the neuroprotective effect of *polyketides*, which are produced by animals, plants, fungi, bacteria, and certain marine organisms [37]. *Terpenoids*, also known as isoprenoids, represent the largest category of natural compounds, accounting for about 60% of known molecules, and they are mostly found in plants, microbial sources, and marine sponges [38]. Numerous evidence has confirmed that terpenoids, such as cannabinoids, have pharmacological properties for treating neurodegenerative diseases [39].

To evaluate natural compounds’ therapeutic effectiveness and potential side effects, human clinical trials have been conducted in recent years. Although hundreds of natural compounds have shown promising effects for treating neurodegenerative disorders in preclinical studies and are proven to be safe in phase I of clinical trials, few molecules have progressed to phase II, III, or IV of clinical trials.

## 4. Natural Compounds in Clinical Trials for Neurodegenerative Diseases

### 4.1. Natural Compounds in Clinical Trials for Alzheimer’s Disease

Natural compounds have been tested in clinical studies for their safety and efficacy in the treatment of AD. Table 1 summarizes the list of natural compounds evaluated in clinical trials for AD treatment.

Vitamins are among the most studied compounds in clinical trials for AD. Their safety and efficacy assessments started in the 1990s and is still ongoing. In particular, vitamin E is the one that has arisen the most interest. This is a fat-soluble vitamin found in several foods, including vegetable oils, cereals, meat, poultry, eggs, and fruits. The in vivo studies have demonstrated the antioxidant and anti-inflammatory properties of this compound and its capability to reduce Aβ levels and amyloid deposition in a transgenic mice model of AD [58]. The New York State Institute for Basic Research (USA) sponsored a randomized, double-blind, placebo-controlled, phase III trial. The trial, lasting 36 months, was conducted on 349 participants to determine the safety and effectiveness of vitamin E on the cognitive function of AD patients with Down syndrome [46]. The participants were randomized to either vitamin E (2000 IU per day) or a placebo. This clinical trial was completed in 2010, but no results have been posted yet.

Galasko et al. (2012) conducted a double-blind, placebo-controlled, phase I clinical trial to determine the effect of the concomitant supplementation of vitamin E with other natural compounds on tau protein and Aβ peptide levels [40]. Over 16 weeks, 78 patients with mild to moderate AD were given a placebo or a daily supplement that contained 800 IU of vitamin E, 500 mg of vitamin C, 900 mg of alpha-lipoic acid, or 1200 mg of coenzyme Q. The researchers concluded that antioxidants had no effect on Aβ or tau levels. The antioxidant group of vitamin E, vitamin C and alpha-lipoic acid showed lower oxidative stress in the brain compared to the placebo group.

Sano et al. (1997) conducted a double-blind, placebo-controlled, randomized, multicenter trial to evaluate vitamin E and selegiline co-therapy in patients with moderate AD [41]. Selegiline is an antioxidant that inhibits oxidative deamination, thus reducing neuronal damage. Over two years, 341 subjects were given daily selegiline (10 mg), vitamin E (2000 IU), selegiline combined with vitamin E (10 mg and 2000 IU, respectively), or a placebo. The co-therapy slowed the time of primary outcomes, including death, institutionalization, and loss of the ability to perform at least two of three basic activities of daily living (eating, grooming, and using the toilet) compared to the control groups.

In turn, Dysken et al. (2014) investigated the co-administration of vitamin E with memantine in a double-blind, placebo-controlled, parallel-group, randomized phase III clinical trial [42]. Memantine is the gold standard drug to control the symptoms of moderate to severe AD [59]. A total of 613 patients with mild to moderate AD were randomly assigned to receive a daily oral dose of placebo, 20 mg of memantine, 2000 IU of vitamin E, or vitamin E combined with memantine for 5 years. Vitamin E was able to slow the functional decline compared with the placebo. However, no improvement was observed in the symptoms of patients receiving memantine or vitamin E combined with memantine.

The benefits of the co-therapy of memantine with another natural compound, vitamin D3, were assessed by Annweiler et al. (2012) in a double-blind, placebo-controlled pilot study involving 43 patients with moderate AD [43]. Vitamin D3 is a natural compound obtained mostly through sunlight exposure with health-promoting properties. The in vivo investigations showed its ability to inhibit the activity of β and γ-secretases, lowering the Aβ production and plaque formation, while increasing Aβ clearance [60]. As a result, improved learning and memory performance was observed in a mouse model of AD [61]. In this clinical trial, memantine (titrated in 5 mg increments over 4 weeks up to a full dose of 20 mg), vitamin D3 (1000 IU), or memantine combined with vitamin D3 were administered daily orally for 6 months. While memantine or vitamin D3 alone did not affect symptoms, the combination of both molecules led to an increase in patients’ cognition. Following that, the same group investigated the co-administration of vitamin D3 with memantine in a phase III unicenter, double-blind, randomized, placebo-controlled, intent-to-treat, superiority trial. Memantine (same dosage) combined with vitamin D3 (100,000 IU every 4 weeks) or a placebo was orally administered to 120 patients with moderate AD for 24 weeks. The clinical trial was completed in 2016 [62], but the results have not yet been published.

B complex vitamins, particularly B6, B9, and B12, are widely believed to have protective effects against AD. The main sources of B vitamins include meat, seafood, and eggs. Aisen et al. (2008) conducted a phase III multicenter, randomized, double-blind controlled clinical trial to determine the efficacy of vitamin B6, B9, and B12 co-supplementation to slow the rate of cognitive decline in mild to moderate AD patients [44]. A total 340 subjects were randomly assigned to receive a daily oral dose of placebo or a combination of vitamin B6 (25 mg), vitamin B9 (5 mg), and vitamin B12 (1 mg) for 18 months. The co-therapy of vitamins B6, B9, and B12 did not improve cognitive performance in AD patients.

Vitamin B3 is another vitamin that was evaluated in a phase II double-blind, placebo-controlled, randomized clinical trial by the University of California (USA). A total of 50 patients with mild to moderate AD received 1500 mg of vitamin B3 or a placebo twice a day for 6 months. The clinical trial was completed in 2014 [45], but the results are yet to be released.

Docosahexaenoic acid (DHA) is the most abundant omega-3 fatty acid found in the human brain and plays a critical role in its development and function [63]. Data from multiple animal models support the concept that DHA, through anti-amyloid, antioxidant, and neuroprotective actions, may be useful in AD therapy [64,65]. A randomized, double-blind, placebo-controlled, phase III trial of DHA supplementation was carried out by Quinn et al. (2010) on 402 patients with mild to moderate AD [47]. The trial was completed by 295 volunteers as 67 and 40 participants from the DHA and placebo group dropped out due to adverse effects. The remaining volunteers took DHA for 18 months at 2 g per day or a placebo. The pilot study concluded that the DHA supplementation did not slow the patients’ cognitive and functional loss compared to the placebo. 

Bryostatin-1, a highly oxygenated macrolide with a polyacetate backbone, is generated by commensal bacteria found in marine invertebrates, primarily bryozoans [66]. Preclinical studies have proven bryostatin to be effective against reduction in Aβ production and enhancing memory and learning in AD mice models [67,68]. Farlow et al. (2019) investigated bryostatin safety, tolerability, and efficacy in improving the cognition of 150 patients with advanced AD [48]. Bryostatin was given to the participants in a double-blind, randomized, placebo-controlled phase II, 12-week experiment. Patients (aged between 55–85) were intravenously injected with 20 µg or 40 µg bryostatin or a placebo daily. This investigation validated the safety of both bryostatin dosages. With 20 µg of bryostatin, cognitive function was improved, unlike the higher dose (40 µg). 

With three phase II clinical trials completed so far by the same company (OPKO Health, USA), scyllo-inositol is another well-studied natural compound for AD. Scyllo-inositol is one of the stereoisomers of inositol, present in dogwood, *Cornus florida* L. (Cornaceae), and coconut palm, *Cocos nucifera* L. (Arecaceae). Preclinical evidence has demonstrated the ability of the compound to inhibit Aβ aggregation and fibril formation, thus improving the memory of an AD mice model [69]. A randomized, double-blind, placebo-controlled, phase II study was performed to explore the safety and efficacy of scyllo-inositol in mild to moderate AD patients [49]. A total of 353 participants was randomly assigned to receive capsules for oral administration containing scyllo-inositol (250 mg) or placebo twice daily for 78 weeks. Although the treatment with scyllo-inositol has shown acceptable safety, the natural compound was unable to provide benefits related to cognitive, functional, and psychiatric impairment. 

Resveratrol is a naturally occurring non-flavonoid polyphenol present in grapes, which has demonstrated several benefits for AD in preclinical studies. In addition to its antioxidant and anti-inflammatory properties, resveratrol was able to prevent tau hyperphosphorylation [70], reduce Aβ fibrillation [71] and induce Aβ clearance [72]. Turner et al. (2015) conducted a randomized, placebo-controlled, double-blind, multicenter phase II trial to examine the safety and tolerability of resveratrol in individuals with mild to moderate AD [50]. The effect of the natural compound on AD biomarkers, including Aβ_1-40_, Aβ_1-42_, and tau, and brain volume loss were also evaluated. A total of 119 subjects received either a placebo or 500 mg resveratrol orally daily over 52 weeks. The dosage was increased by 500 mg every 13 weeks, with a total increase of 2000 mg. Aside from causing nausea, diarrhea, and weight loss, resveratrol did not alter AD biomarker levels. In addition, resveratrol caused a significant reduction in brain volume compared to the placebo.

The co-administration of resveratrol with other molecules was recently investigated in a randomized, double-blind, placebo-controlled phase III trial by Zhu et al. (2018) [51]. The group assessed the safety, tolerability, and efficacy of a mixture containing 5 mg of resveratrol, 5 g of glucose, and 5 g of malate to slow AD progression. The mixture or placebo was administered twice a day to 39 patients diagnosed with mild to moderate AD for 1 year. Although the preparation was safe and well tolerated, it was ineffective in slowing the progression of the disease.

Curcumin is another natural compound whose safety and therapeutic efficacy for AD has been investigated in clinical trials. Curcumin is an antioxidant and anti-inflammatory molecule present in the root of *Curcuma longa* L. (Zingiberaceae). This compound has the ability to reduce β and γ-secretase levels, inhibit Aβ aggregation and disaggregate Aβ fibrils [73]. As a result, the spatial learning and memory deficits of the AD rat model were improved [74]. Ringman et al. (2012) conducted a randomized, double-blind, placebo-controlled phase II clinical trial to evaluate the safety and efficacy of curcumin in AD patients [52]. For 24 weeks, 36 subjects with mild to moderate AD were randomly assigned to receive either an oral placebo or 4 g of curcumin daily. Although curcumin was mostly well accepted, three participants withdrew due to gastrointestinal complications. Moreover, the authors were unable to prove the clinical efficacy of curcumin as no differences between the two groups in cognitive performance and biomarker levels (Aβ_1-40_, Aβ_1-42_, and tau) were observed. 

Curcumin was also studied in combination with Ginkgo extract, an extract rich in flavonoids and terpenoids, or a placebo combined with Ginkgo extract. This clinical study was carried out by Baum et al. (2008) to study the beneficial effects of this combination in slowing down AD progression [53]. The authors carried out a randomized, double-blinded, and placebo-controlled phase II study on 34 AD patients over 6 months. The patients took 1 or 4 g of curcumin per day (through capsules or powdered form) combined with 120 mg per day of Ginkgo extract or a placebo combined with Ginkgo extract. They concluded that curcumin slows AD development by lowering oxidation in AD patients. However, some adverse events in AD patients with 4 g curcumin dosage were observed. 

Huperzine A, an acetylcholinesterase inhibitor derived from the fir moss *Huperzia Serrata*, has been used in traditional Chinese medicine for fever and inflammation in a variety of illnesses. With a suspected mechanism of action via central nicotinic and muscarinic receptors, it has been clinically used for AD treatment and other kinds of dementia [75]. A randomized, placebo-controlled phase II clinical study conducted by Rafii et al. (2011) investigated the safety and effectiveness of huperzine A on patients with mild to moderate AD [54]. In total, 210 people were randomized to receive a twice a day placebo or huperzine A (200 µg or 400 µg) for at least 16 weeks. With huperzine A 200 µg, the primary efficacy analysis failed to demonstrate any cognitive advantage after 16 weeks, whereas the 400 µg dose showed evidence of cognitive enhancement. It was well tolerated at both tested doses. 

Homotaurine, also called tramiprosate, generally found in algae, is a blood–brain barrier (BBB)-permeable amino acid that antagonizes amyloid fibril formation [76]. In vitro and in vivo, homotaurine protects against Aβ-induced neurotoxicity in neuronal and mouse organotypic hippocampal cultures, and it reverses Aβ-induced long-term potentiation inhibition in the rat hippocampus [77]. Further research revealed that oral homotaurine could cross the BBB and protect the brains of transgenic mice that overexpressed human amyloid protein from developing amyloid plaques [78]. Based on these observations, Aisen et al. (2006) conducted a randomized, double-blind, placebo-controlled phase II study on 58 participants with mild to moderate AD. The participants were randomly assigned to receive a placebo or homotaurine 50, 100, or 150 mg (two times a day) for 3 months. Following the completion of the double-blind phase, 42 of these participants were put into an open-label phase in which they received 150 mg (two times a day) of homotaurine for 17 months. They concluded that homotaurine was safe and tolerable and reduced Aβ_42_ levels in the cerebrospinal fluid. With the success of the phase II clinical study, the same group also conducted a phase III double-blind, placebo-controlled trial on 1052 patients with mild to moderate AD symptoms (who were taking stable doses of cholinesterase inhibitors alone or in combination with memantine). Patients were randomized to homotaurine 100 mg, 150 mg, or placebo for 78 consecutive weeks, taking two doses daily. The study concluded that homotaurine (150 mg) has an effect in slowing hippocampal atrophy and showed evidence of a beneficial effect on cognition [55,79].

Melatonin is another natural compound that has been evaluated in a phase II double-blind, parallel-group, randomized, placebo-controlled clinical trial. It is an antioxidant and anti-inflammatory molecule found in animals, plants, fungi, and bacteria. Melatonin inhibits the β- and γ-secretase activities while increasing the α-secretase activity, thus preventing Aβ production [80]. Wade et al. (2014) evaluated the effect of melatonin tablets (2 mg) on the cognitive functioning of 80 patients diagnosed with mild to moderate AD [56]. Melatonin or placebo tablets were administered daily for 24 weeks. Melatonin-treated patients showed an improvement in cognitive performance compared to placebo-treated patients.

EGCG is a polyphenol with antioxidant and anti-inflammatory properties found in green tea that has shown neuroprotective effects in AD animal models [81]. EGCG is able to reduce β- and γ-secretase activities while increasing the α-secretase activity [82]. Furthermore, it inhibits Aβ and tau aggregation [83] and increases phosphorylated tau clearance [84]. The therapeutic efficacy of EGCG has also been studied in a phase III parallel-group, randomized, placebo-controlled clinical trial by Charite University (Germany). Increasing doses of EGCG (200 mg to 800 mg) were administered to 21 AD patients in the early stage of the disease for 18 months. Although the trial was completed in 2015 [57], the outcomes have not yet been published.

### 4.2. Natural Compounds in Clinical Trials for Parkinson’s Disease

Several clinical trials have recently explored natural products’ roles in PD treatment. Those studies are summarized in Table 2.

With four published studies, nicotine has received the greatest attention among all the natural compounds in clinical trials for PD therapy. Nicotine is reported to have a neuroprotective effect and to improve some motor and cognitive symptoms [94]. These clinical trials were designed to address PD symptoms, and all were completed [85,95,96,97]. For instance, Lieberman et al. (2019) investigated the treatment of motor impairments in PD using nicotine in a randomized, single-center, double-blind, placebo-controlled phase I/II trial [87]. Participants with idiopathic PD (*n* = 65) were separated into two groups and administered an oral placebo or oral capsules in rising doses of 1 mg to 6 mg of nicotine four times per day for 10 weeks. Nicotine significantly reduced falls and gait freezing, two motor symptoms of PD, compared to the placebo group. No difference in dyskinesias between groups was observed.

Caffeine is another common natural compound with relevant neuroprotective, motor, and cognitive benefits in PD treatment, acting by different mechanisms. Aside from that, genetic studies have provided essential insights into the interactions between caffeine and several genes linked to PD pathogenesis [98]. Postuma et al. (2017) explored caffeine as a therapeutic agent in a phase III study on 121 patients with idiopathic PD [86]. In this randomized, multicenter, double-blind, placebo-controlled trial, the participants received 200 mg of caffeine in capsules or a placebo twice daily for 6–18 months. The authors found a slight decline in cognitive test scores and alertness improvement in patients receiving caffeine. However, an overall rise in dyskinesia was observed. Previously, the same researchers conducted two other clinical studies evaluating caffeine’s role in non-motor and motor PD symptoms [99,100,101]. Contrary to the findings of this phase III study, the other clinical studies revealed that caffeine may have some motor benefits. However, the experiments were carried out with fewer participants and for a shorter time period. Therefore, caffeine can provide a short-term advantage that disappears fast. These investigations mainly concluded that caffeine could not be used for PD symptomatic therapy.

Cannabidiol is a natural compound that has been reported to reduce tremors, anxiety, and psychosis in PD patients [87,102].The cannabidiol impact on tremors of idiopathic PD patients was investigated by Leehey et al. (2020) in a randomized, multicenter, double-blind, placebo-controlled, phase II study with 13 participants [87]. Cannabidiol was administered as an oral solution for up to 5 weeks. The dosage started at 5 mg/kg/day and was increased by 2.5–5 mg/kg at 3–5-day intervals until it reached a target dose of 20 mg/kg/day. The key study variables, dyskinesia and tremor, showed significant improvements. The same group also completed a phase II study of the tolerability and efficacy of cannabidiol on motor symptoms in PD in February 2022, with no published results yet [103]. 

Other potential approaches for PD treatment include vitamins. According to recent clinical research, vitamins may help treat PD due to their antioxidant properties and influence on gene expression regulation [104]. Wakade et al. (2021) evaluated the safety and tolerability of vitamin B3 in early- and late-stage PD patients and whether vitamin B3 supplementation may reduce inflammation and improve their symptoms. The study consisted of a randomized, single-center, double-blind, placebo-controlled trial with 47 participants [89]. For 6 months, the participants received vitamin B3 (250 mg tablets) or placebo. Vitamin B3 supplementation significantly reduced fatigue, improved motor functions, and regulated serotonin levels, which could help in non-motor symptoms. 

DiFrancisco-Donoghue et al. (2012) studied the benefit of co-supplementation with vitamins B6, B9 and B12, combined or not with physical activity, in homocysteine and glutathione levels, two markers associated with PD [88]. This open-label clinical trial was conducted for 6 weeks with 40 participants that were randomly divided into four groups. The first group received vitamins (25 mg/day of vitamin B6, 5 mg/day of vitamin B9, and 2000 µg/day of vitamin B12), the second group received training twice weekly, and the third group received both under the same conditions. The fourth group was the control, which received no interventions. The participants who received vitamin supplementation had lower homocysteine levels, whereas those who underwent training showed higher glutathione levels. The combination of training and supplementation did not have more significant effects than either of the strategy alone. 

Vitamin D3, for example, is reported to enhance motor function in PD [104]. Hiller et al. (2018) assessed the impact of vitamin D3 supplementation on balance and other motor and non-motor features of PD in a randomized, double-blind, placebo-controlled trial [90]. In this phase II study, the participants (*n* = 101) were given either a high vitamin D supplementation (10,000 IU/day) or a placebo for 16 weeks. Additionally, both groups received 1 g of calcium once daily. No improvement in motor and non-motor parameters was observed after vitamin D administration. The authors concluded that more research with high dosage vitamin D supplementation in PD is needed, as a post hoc analysis revealed that vitamin D might improve balance in the younger participants with PD. 

Vitamin E plays a role in various physiological functions, including regulating gene expression and cognitive and motor functions. This vitamin is a powerful antioxidant with neuroprotective benefits shown in many experimental models and may help treat the PD illness [104]. The combined effect of vitamin E and coenzyme Q10 in early PD was evaluated by Beal et al. (2014) in a randomized, double-blind, placebo-controlled, phase III clinical trial [91]. This study consisted of 600 participants randomly treated for 16 months with 1200 IU/day of vitamin E and 1200 mg/day or 2400 mg/day of coenzyme Q10. Compared to the placebo group, none of the therapy groups showed any advantage in PD treatment or prevention. 

DHA is reported to restore dopamine levels and protect motor function in PD [105]. Its impact on reducing dyskinesia in PD was investigated in March 2012 by Kathryn Anne Chung from the Veterans Affairs Portland Health Care System (USA) in collaboration with the Oregon Health and Science University (USA) [92]. This randomized, multicenter, triple-blind, placebo-controlled, phase I study was conducted on 33 participants, divided into two groups, receiving a daily dose of 2 g of DHA or placebo for 18 months. High DHA levels were found in both plasma and cerebrospinal fluid without significant side effects, demonstrating the compound’s safety and tolerability. The authors also confirmed that patients receiving DHA had higher levels of absent dyskinesia than the control group. 

In PD, EGCG is reported to improve motor impairments and offer neuroprotection by preventing neuroinflammation and neurodegeneration [106]. Despite the promising properties demonstrated in vitro and in vivo [107], only a single clinical trial exists using EGCG in PD management. Chan et al. (2009) conducted a randomized, double-blind, placebo-controlled, phase II study to assess the neuroprotection effect of EGCG in de novo PD patients [93]. De novo PD patients are either newly diagnosed patients with PD or not receiving any treatment. The participants (*n* = 480) received EGCG (0.4, 0.8, or 1.2 g daily given in two equal oral doses) or a placebo. Patients were treated for one year, and after 6 months, the placebo was switched to 1.2 g daily of EGCG. The rating method used to assess the progression of PD showed a significant improvement in the treatment groups over the placebo group after 6 months. The results were no longer significantly different, though, after a year. Although EGCG helped PD patients with their symptoms, the authors concluded that it did not appear to have any noticeable disease-modifying effects.

### 4.3. Natural Compounds in Clinical Trials for Multiple Sclerosis

Natural compounds have been evaluated in clinical trials in order to assess their safety and efficacy for the treatment of MS. Table 3 has summarized the list of natural compounds with completed or active clinical trials for MS treatment.

Vitamin D3 is undoubtedly the most explored natural product in clinical trials for MS therapy since recent evidence has shown a correlation between low vitamin D serum levels and an increased risk for MS [121]. Of 18 registered studies, 3 are currently in progress or under the recruitment stage [122,123,124], 3 were terminated [125,126,127], and 9 were completed. Among the completed studies, two phase III clinical trials [128,129] and one phase IV clinical trial [130] were conducted, but the results are not yet publicly available. Most recently, Camu et al. (2019) designed a phase II randomized, double-blind, placebo-controlled trial with 129 relapsing–remitting MS patients aged 18 to 65 [108]. In this study, the intervention group received a daily vitamin D3 dietary supplement of 100,000 IU for 96 weeks. The primary outcome, to reduce the relapse rate, was not met. However, the results suggested a potential beneficial effect of supplementation with the vitamin, with the treated patients showing less axonal loss and matrix destruction.

Other vitamins, such as vitamin A, have shown the ability to ameliorate MS symptoms by reducing the inflammatory response in the CNS [131]. Additionally, evidence has shown lower blood levels of vitamin A in MS patients, leading to the dysregulation of immune tolerance and to the pathogenic immune cell production in the CNS [132]. Thus, in the last years, the Tehran University of Medical Sciences (Iran) sponsored four different phase IV clinical trials to evaluate the effect of this vitamin in MS patients. In 2011, they started a clinical trial with a higher participant number (*n* = 100). In this double-blind, placebo-controlled trial, MS patients aged between 20 and 45 were divided in two groups, placebo and intervention group. For the first six months, the patients received a daily oral supplement of 25,000 IU of vitamin A and 10,000 IU for the next 6 months. The results showed that, although vitamin A supplementation ameliorated symptoms such as fatigue and improved psychiatric outcomes such as depression [109], it did not change the disability status, relapse rate, and brain active lesions [110].

Alpha-lipoic acid is a natural compound widely found in animal products such as red and organ meats with antioxidant properties. This molecule has multiple biological functions, and so far, it is the bioactive compound that possesses the most robust evidence for efficacy in MS [133]. The in vitro and in vivo studies revealed that alpha-lipoic acid can protect neuron cells by inhibiting the expression of inflammatory mediators and by decreasing the immune cell infiltration in the CNS [134]. Furthermore, in clinical studies, the compound improved the patients’ walking performance, gait, and balance. For these reasons, alpha-lipoic acid is one of the most studied natural compounds in clinical trials for MS therapy, being evaluated alone or in combination in seven studies in the last decade. The first study to investigate the effect of this natural compound (alone) in MS patients was a phase I trial conducted in 2005 [135], and after that, two other phase I trials [136,137] and three phase II trials were conducted [138,139,140]. Most recently, Spain et al. (2017) evaluated the clinical benefits of alpha-lipoic acid supplementation in patients (*n* = 54) with secondary progressive MS aged between 40 and 70 in a single-center, double-blind, placebo-controlled phase III clinical trial [111]. To evaluate if this compound can slow disability and protect the brain, the patients received a placebo or 1.2 g of alpha-lipoic acid per day by oral administration for 2 years. The study proved that this natural compound prevents brain volume loss while improving the symptoms of MS. Additionally, the administration of alpha-lipoic acid has been proven to be safe and tolerable, with high patient compliance under this studied dose. 

The therapeutic effect of alpha-lipoic acid was also evaluated in combination with other natural compounds in two clinical trials. In 1999, a phase II clinical trial was sponsored by the National Center for Complementary and Integrative Health (USA) aiming to evaluate the effect of antioxidant therapy in decreasing the markers of oxidative damage in MS patients [112]. A mixture of alpha-lipoic acid, Ginkgo biloba, fatty acids, vitamin E, and selenium was administered to the patients. The study was completed in 2004, but no results were posted. 

A few years later, in 2014, the Oregon Health and Science University (USA), together with the National Multiple Sclerosis Society (USA), conducted a phase II clinical trial to evaluate if the combined supplementation with alpha-lipoic acid and omega-3 fatty acids could improve the cognitive function in MS patients [113]. The authors designed a randomized, double-blind, placebo-controlled pilot trial with 54 MS patients aged between 18 and 65 with cognitive impairment. The intervention group was administered orally with 1.2 g per day of alpha-lipoic acid, 1.35 g of DHA, and 1.95 g of eicosapentaenoic acid (EPA) for 12 weeks. Twelve of the fifty-four patients did not complete the trial due to adverse effects (*n* = 9) or other reasons (*n* = 3). No significant changes were verified in the patients’ memory, attention, verbal learning, and letter fluency, whether they were receiving a placebo or the natural compound mixture. 

Omega-3 polyunsaturated fatty acids have shown potential for MS therapy by reducing inflammatory markers, glutathione reductase, and the relapse rate [141]. Thus, the potential of these omega-3 polyunsaturated fatty acids for MS therapy was also evaluated in other clinical trials. Ramirez-Ramirez et al. (2013) conducted a phase IV randomized, double-blind, placebo-controlled trial on patients with relapsing–remitting MS (*n* = 50) [114]. The patients were divided into two groups and received a placebo or oral capsules containing 1.6 g of DHA and 0.8 g of EPA for 12 months. The findings revealed that, although the mixture led to a decrease in the inflammatory cytokines, the treatment regimen did not improve clinical efficacy regarding the expanded disability status scale and the annualized relapse rate. 

Another natural compound, melatonin, has also been explored as a therapeutic target in different clinical trials for MS, since recent studies have shown that melatonin secretion is dysregulated in MS patients [142]. Among three registered studies, two are currently in progress [143,144] and one was completed [115]. Roostaei et al. (2015) led a double-blind, randomized, parallel-group, placebo-controlled phase II clinical trial [115]. In this study, relapsing–remitting MS patients aged 20 to 45 (*n* = 25) were administered 3 mg/day of oral melatonin in comparison to a placebo for 12 months. The treatment proved to be safe, with no patients experiencing adverse effects. However, no significant differences were verified between the placebo and the treatment groups regarding the number of relapses, brain volume changes, and lesions. 

The anti-inflammatory agent EGCG has shown neuroprotective effects in animal models of MS [145], and, therefore, it has also evaluated as a potential therapeutic molecule for MS treatment. Currently, there are seven registered clinical trials to assess the effect of EGCG or EGCG-containing extracts in MS patients. Among them, only one has evaluated the effect of the natural compound in its pure form [116]. This clinical trial started in 2018 and it was sponsored by the Fundación Universidad Católica de Valencia San Vicente Mártir (Spain). In this phase II randomized placebo-controlled trial, 60 MS patients aged 19 to 65 were divided into a placebo and treatment group, and the latter was orally administered with 600 mg of EGCG and 60 mL of coconut oil per day, divided into two doses for 4 months. The trial started in 2018 and ended in 2019. The results proved that supplementation with the natural compounds was beneficial for MS patients by decreasing depression levels [119], markers of inflammation [117], and cardiovascular risk [118].

Curcumin has also been explored for MS treatment due to its well-reported anti-inflammatory properties. Data have shown that curcumin can inhibit the secretion of proinflammatory cytokines, preventing the cascade of inflammation involved in neuronal damage [146]. Petracca et al. (2021) assessed the efficacy, safety, and tolerability of dietary supplementation of curcumin in patients with active relapsing MS [120]. The authors designed a randomized, double-blind, placebo-controlled phase II trial with 41 participants. The patients were given a placebo or 500 mg of curcumin orally twice a day for 24 months. For the entire experiment duration, both groups were administered subcutaneously with 44 µg of interferon β-1a, and immunomodulators that are used to treat the relapsing forms of MS. Despite the verified high drop-out rate, with 22 participants leaving the trial, the authors still concluded that while curcumin decreased the radiological signs of inflammation, it did not exert any neuroprotective effects.

### 4.4. Natural Compounds in Clinical Trials for Amyotrophic Lateral Sclerosis

Some clinical trials using natural compounds have reported their safety, tolerability, and potential to treat ALS. So far, just four clinical trials were completed. Some others are still under evaluation. Table 4 summarizes the clinical trials using natural compounds for ALS treatment that are either completed, active, or recruiting patients.

Nowadays, vitamins are the most studied compounds in clinical trials for the treatment of ALS. Among other vitamins, vitamins B and E, have been reported to be able to prevent and treat ALS symptoms in vivo [156]. MedDay Pharmaceuticals AS (France) led a clinical trial to evaluate the vitamin B7′s effect and safety. The company did a quadruple-blinded, placebo-controlled, randomized trial where participants, care providers, investigators, and outcomes assessors were blinded to the allocation status of the participants. This was a phase II trial with 30 participants between 25 to 80 years, including male and female ones, with probable or confirmed ALS. The patients were treated orally with capsules containing 100 mg of vitamin B7 three times per day for 12 months. The bioactive compound demonstrated a safety profile and was well tolerated by the patients [147]. Due to the low number of participants (20 received the drug and 10 the placebo), it was impossible to prove the treatment effectiveness. Another trial with 12 patients diagnosed within 3 years prior to participation with possible, probable, or definite ALS using this vitamin was performed; however, the results are not reported yet [157]. 

Another trial using methylcobalamin (the activated form of vitamin B12) was conducted with 373 male and female participants older than 20 years and diagnosed with probable or confirmed ALS. It was a randomized, double-blind, placebo-controlled experiment in phases II and III. The participants who received the therapeutic compound were divided into two groups, one receiving 25 mg of methylcobalamin twice a week for 3.5 years and the other receiving 50 mg of methylcobalamin twice a week for 3.5 years. Both routes of administration were intramuscular. Kaji, R. and co-workers (2015) reported that methylcobalamin significantly prolonged the survival and retarded the disease progression in both studied conditions [148]. 

Vitamin E’s therapeutic efficacy was also evaluated in a clinical trial phase III sponsored by Lawson Health Research Institute (United Kingdom, UK) started in 2006 [150] with 32 people aged 18 or older, male or female, with probable or definite ALS. It was a randomized, quadruple-blind trial, with a crossover assignment experiment, with all groups of patients receiving vitamin E 800 IU twice a day. Arm one of the experiment received vitamin E first and placebo second. Arm two received the placebo first, then vitamin E. This trial was completed in 2016, however, the results are still unknown. 

Another approach to utilize vitamins to treat ALS is to combine them with other molecules, expecting a synergic effect between them. A clinical trial (phase II) designed by Dallas VA Medical Center (USA) that will use a mixture of vitamin E, NAc cysteine, L-cystine, nicotinamide, and taurursodiol is now recruiting participants [151]. 

Phoenix Neurological Associates LTD (USA) started a trial (phase I and II) in 2008 to test the effect of tretinoin (also known as all-trans retinoic acid, an active metabolite of vitamin A) in combination with pioglitazone hydrochloride (an anti-diabetic drug for the treatment of Type 2 diabetes) [152]. The researchers expected a significant delay in the disease progression with the use of these two Food and Drug Administration (FDA)-approved molecules in combination. It was a randomized trial, triple-blinded, with a parallel assignment. The patients with ALS (aged between 18 and 85) were exposed to one pill of pioglitazone hydrochloride (10 mg) twice a day and one pill of tretinoin once a day (30 mg). The safety and efficacy of this treatment were evaluated over 3 years; however, no results are available.

Another molecule tested in humans for treating ALS was theracurmin (a colloidal dispersion of curcumin) [153]. Curcumin has the capacity to improve motor function and survival time in a mouse model of spinal and bulbar muscular atrophy, a motor neuron disease commonly confused with ALS [158,159]. Based on its beneficial properties, a phase II clinical trial started in 2020 with 68 participants, coordinated by Richard Bedlack from Duke University (UK). It was a non-randomized, open-level trial, with a parallel assignment experiment. In this trial, ALS patients older than 18 years with mild cognitive impairment received 90 mg of theracurmin in capsules twice a day. The results of this trial have not been published yet.

Apart from vitamins and theracurmin, some other natural compounds and plants with antioxidant and anti-inflammatory properties, such as nicotinamide (a water-soluble form of vitamin B3) and cannabis, will be studied in humans soon [149,155,160]. Specially, cannabis has revealed outstanding anti-excitotoxicity, antioxidant and anti-inflammatory properties. In transgenic murine with ALS, it was proven that the administration of cannabinoids increases survival time and decreases disease progression [161]. Encouraged by those findings and previous trials, a clinical trial using cannabis is already in ongoing phase IV, with 200,000 participants [155]. 

Another alternative is the co-administration of natural compounds encapsulated in liposomes. Since one of the main problems related to natural compounds is their low bioavailability, using nanoparticles (NPs) to transport these molecules without modifying their physicochemical properties could be an interesting approach to overcome those limitations. José Enrique de la Rubia Ortí, from Fundación Universidad Católica de Valencia San Vicente Mártir (Spain), is coordinating a clinical trial using a combination of resveratrol and curcumin encapsulated in liposomes [154]. In transgenic ALS animals, resveratrol significantly delayed the disease onset, preserves the motoneuron function, and consequently increases the survival time [162]. Aside from the natural compounds being protected by the NPs, a synergic effect is expected using these two molecules, with a high potential to treat the ALS. The trial is expected to start in November 2022, and it is still recruiting patients.

### 4.5. Natural Compounds in Clinical Trials for Huntington’s Disease

Natural compounds have been evaluated in clinical trials to assess their safety and efficacy for the treatment of HD. Table 5 summarizes the list of natural compounds with completed or active clinical trials for HD treatment.

Cannabidiol has shown neuroprotective effects and has attracted interest as a therapeutic molecule to slow down HD progression. This compound can prevent oxidative stress by downregulating the expression of superoxide-generating enzymes and prevent neuroinflammation by reducing the expression of inflammatory mediators [168]. Moreno et al. (2016) studied the efficacy of cannabidiol on HD symptoms in a double-blind, randomized, placebo-controlled phase II trial [163]. In this study, 18 years old or older HD patients (*n* = 24) received the natural compound in combination with another cannabinoid (delta-9-tetrahydrocannabinol, THC) as an oral spray (one spray per day, up to a maximum of 12 sprays per day) for 12 weeks. The results did not show a relevant improvement in the patients’ symptoms, and the authors concluded that future studies should consider higher doses and longer regimens.

EGCG was also investigated as a therapeutic approach for HD, since it can prevent early events in the aggregation process by modulating the misfolding and oligomerization of the mutant htt exon 1 protein in vitro [168]. The efficacy and tolerability of EGCG supplementation were evaluated in 2011 in a double-blind, randomized, placebo-controlled phase II trial with 54 participants sponsored by the Charite University (Germany) [164]. The patients (18 years or older) were randomly divided and received a placebo or an increasing oral dosage of EGCG (from 400 mg/day to 1200 mg/day) for 3 months. After the first 3 months, the patients received the higher dosage (1200 mg/day) for an additional 9 months. The trial lasted from 2011 to 2015, but no results are posted yet.

Resveratrol is another natural compound that has shown therapeutic potential for HD. In an animal model of the disease, this compound reduced mitochondrial dysfunction related to motor disturbances [169]. The potential of resveratrol for HD therapy was studied in 2015 in a randomized, double-blind, placebo-controlled trial sponsored by the Hôpitaux de Paris (France) [165]. The trial involved 102 participants administered oral capsules containing a placebo or resveratrol 80 mg/day every day for 1 year. The trial was ended in 2020, and no results are yet posted.

## 5. Discussion and Concluding Remarks

Despite the substantial advances in understanding the pathology of neurodegenerative diseases, developing new effective treatments has been challenging. Considering the socioeconomic burden of neurodegenerative diseases and the adverse drug side effects that lead to low patients’ compliance, new therapeutic strategies are required. As a result, using natural compounds has been recently assessed as a new therapeutic approach with a significant pharmacological impact and minimal side effects. 

This review highlighted the therapeutic potential of several natural compounds on the pathologies of the five more prevalent neurodegenerative diseases. Fifty natural compounds or combinations of natural compounds in clinical trials for neurodegenerative diseases were found and discussed. Although the majority of these studies use a single natural compound, others combine two or more. Even though each of the natural compounds has promising benefits for the therapy of neurodegenerative diseases on its own, when combined, a synergetic therapeutic impact may be created in which the whole effect may exceed the sum of the individual effects. Through the studies, one can find that most of the reported clinical trials with natural compounds focused on treating AD, accounting for 18 natural products being studied in total trials (corresponding to 36% of the studies), as illustrated in Figure 1. For PD, MS, and ALS only nine natural compounds or a combination of natural compounds were evaluated in clinical trials (18% for each disease), whereas for HD only five natural products were assessed in human studies (equivalent to 10%).

Vitamin B alone and in combination with other natural compounds received the most attention in clinical trials for the specified neurodegenerative diseases (accounting for eight clinical studies), followed by Vitamin E in combination with other natural compounds, with a total of six clinical studies. Due to their antioxidant and anti-inflammatory properties, vitamins are being widely studied for the treatment of these five neurodegenerative diseases. 

From the herein discussed clinical trials, 4% involved natural compounds in a clinical phase trial I, 48% of compounds were featured in a clinical phase trial II and 22% involved phase III clinical trials. Only 6%, corresponding to three clinical trials, reached the clinical phase trial IV. The two natural compounds that reached the most advanced clinical phase trial, phase IV, were vitamin A and a combination of DHA and EPA for MS therapy [109,110,114]. The results were unsatisfactory for the DHA-EPA combination, and the treatment regimen did not demonstrate improved clinical efficacy. The authors noted that one of the primary drawbacks of the study was the small sample size, which was performed on only 50 participants. A phase IV clinical trial for the treatment of ALS will also assess the long-term tolerability and safety of cannabidiol [154]. For this study, recruiting has not yet begun. The remaining 14% are clinical trials that do not include phase information.

Despite the encouraged in vitro and in vivo outcomes [170,171,172,173], when translated to clinical trials, some natural compounds did not show the expected significant therapeutic effects [44,47,49,50,51,52,91,113,120,163]. This lack of efficacy in clinical trials may be attributable to the physiological differences between animals and humans and the trials’ inherent limitations, including their short duration, reduced number of subjects, and impact of genetic, demographic, and clinical factors on subjects’ outcomes [79]. The complex pathogenic pathways and the several underlying mechanisms for the onset and progression of these neurodegenerative diseases may also be to blame for the failure of the clinical studies [174]. For AD, for instance, there is a growing evidence that suggests the illness may be brought on by a combination of mechanisms rather than by one single cause [175]. In this approach, identifying a natural compound that might be effective for all underlying mechanisms and affected disease targets may not be possible, and therefore, a therapy based on a combination of natural compounds as a multi-target strategy may be more promising or preferable than a therapy including just one natural compound.

Clinical trials may also fail due to the inherent pharmacokinetic and pharmacodynamic features of natural compounds that compromise their therapeutic efficacy. These include poor bioavailability, limited stability, susceptibility to physical and chemical degradation, and rapid metabolism. Besides, the BBB, a biological membrane that restricts the natural compounds’ access to the target tissue—the brain—is another huge challenge that compromises the neuroprotective effects of natural compounds. For neurodegenerative diseases, the ability of natural compounds to cross the BBB is crucial. In this way, nanotechnology might play a vital role to promote the natural compounds’ access to the brain and improve their therapeutic role. NPs can increase the therapeutic effectiveness of therapeutic molecules by protecting them and increasing their bioavailability. Their surface could also be modified for targeted brain administration, enabling the compounds’ release into the desired tissue [176,177]. In fact, NPs are used to improve natural compounds’ effectiveness in one clinical trial. This trial is recruiting participants and is anticipated to begin in November 2022. The sponsors will assess the liposomal encapsulation of resveratrol and curcumin for ALS treatment by combining the natural compounds’ synergistic effects with NPs’ protection of the natural compounds [154]. 

Despite being regarded as a safe strategy, the clinical use of natural compounds to treat neurodegenerative diseases is still uncommon due to the lack of enough clinical studies supporting their efficacy. As a result, before natural compounds may be approved as pharmacological approaches, future studies are required to understand how they manifest their therapeutic effects and to find novel ways to deliver them to the brain. 

## Figures and Tables

**Figure 1 pharmaceutics-15-00212-f001:**
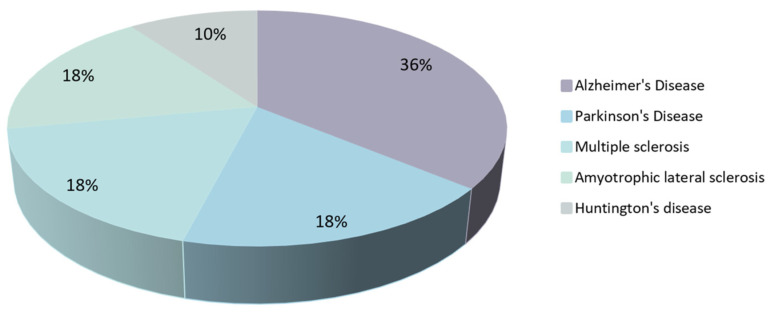
Schematic distribution of the number of natural compounds studied in clinical trials by the targeted disease. The graph was created based on the clinical trials reported in this review.

**Table 1 pharmaceutics-15-00212-t001:** Natural compounds in clinical trials for AD therapy.

Natural Compound	Condition	Nº of Subjects	Duration	Phase	Main Outcomes	Status	Starting Year	Ref.
Vitamin E, vitamin C, alpha-lipoic acid or coenzyme Q	AD	75	16 weeks	I	Vitamin E, vitamin C and alpha-lipoic acid group had shown less oxidative stress in the brain	Completed	2006	[40]
Vitamin E and selegiline	Moderate AD	341	2 years	n.d.	Delay of symptoms and patients’ death	Completed	n.d.	[41]
Vitamin E and memantine	Mild to moderate AD	613	5 years	III	Vitamin E slows functional decline. Memantine alone and vitamin E with memantine without effect	Completed	2007	[42]
Vitamin D3 and memantine	Moderate AD	43	6 months	Pilot study	Improvement of patients’ cognition	Completed	2011	[43]
Vitamin B9, B6, and B12	Mild to moderate AD	340	18 weeks	III	No effect	Completed	2003	[44]
Vitamin B3	Mild to moderate AD	50	24 weeks	II	No results posted	Completed	2008	[45]
Vitamin E	AD with Down syndrome	349	36 months	III	No results posted	Completed	2000	[46]
DHA	Mild to moderate AD	295	18 months	III	No effect	Completed	2007	[47]
Bryostatin	Severe AD	150	12 weeks	II	Improvement in cognitive function with lower dose; Safe and well-tolerated	Completed	2015	[48]
Scyllo-inositol	Mild to moderate AD	353	78 weeks	II	Acceptable safety; No therapeutic effects	Completed	2007	[49]
Resveratrol	Mild to moderate AD	119	52 weeks	II	Side effects; No therapeutic effects	Completed	2012	[50]
Resveratrol, glucose, and malate	Mild to moderate AD	39	1 year	III	Safe and well tolerated; No therapeutic effects	Completed	2007	[51]
Curcumin	Mild to moderate AD	36	24 weeks	II	Mostly well tolerated; No therapeutic effects	Completed	2003	[52]
Curcumin and Ginkgo extract	AD	34	24 weeks	II	Lower oxidative stress in the brain	Completed	2004	[53]
Huperzine A	Mild to moderate AD	177	16 weeks	II	Cognitive enhancement with higher dose; Safe and well tolerated at both doses	Completed	2004	[54]
Homotaurine	Mild to moderate AD	1052	78 weeks	III	Slowing hippocampal atrophy	n.d	2004	[55]
Melatonin	Mild to moderate AD	80	24 weeks	II	Improvement in cognitive performance	Completed	2009	[56]
EGCG	Early stage of AD	21	18 months	III	No results posted	Completed	2009	[57]

n.d.—not disclosed.

**Table 2 pharmaceutics-15-00212-t002:** Natural compounds in clinical trials for PD therapy.

Natural Compound	Condition	Nº of Subjects	Duration	Phase	Main Outcomes	Status	Starting Year	Ref.
Nicotine	Idiopathic PD	65	14 weeks	I and II	Improvement of motor symptoms	Completed	2009	[85]
Caffeine	Idiopathic PD	119	18 months	III	Decline in cognitive functions and improvement of alertness	Completed	2012	[86]
Cannabidiol	Idiopathic PD	13	5 weeks	II	Improvement in several motor and non-motor symptoms	Completed	2016	[87]
Vitamins B6, B9, and B12	PD	40	6 weeks	n.d.	Improvement of homocysteine and glutathione levels	Completed	2008	[88]
Vitamin B3	Early- and late-stage PD	47	6 months	n.d.	Improvement of motor impairment; reduction of neuroinflammation	Completed	2016	[89]
Vitamin D	PD	101	16 weeks	II	Improvement of PD patient’s balance	Completed	2010	[90]
Vitamin E with coenzyme Q10	Early PD	600	16 months	III	No therapeutic effect	Completed	2008	[91]
DHA	PD	33	1.5 years	I	Reduced dyskinesia	Completed	2018	[92]
EGCG	de-novo PD	480	1 year	II	Slight clinical benefit; Well tolerated	Completed	2006	[93]

n.d.—not disclosed.

**Table 3 pharmaceutics-15-00212-t003:** Natural compounds in clinical trials for MS therapy.

Natural Compound	Condition	Nº of Subjects	Duration	Phase	Main Outcomes	Status	Starting Year	Ref.
Vitamin D3	Relapsing-remitting MS	129	96 weeks	II	No changes in the relapse rate; Less axonal loss and matrix destruction	Completed	2009	[108]
Vitamin A	Relapsing-remitting MS	100	1 year	IV	No changes in the relapse rate and brain lesions but ameliorated symptoms	Completed	2011	[109,110]
Lipoic acid	MS	54	2 years	III	Prevention of brain volume loss and amelioration of symptoms; Safe and well tolerated	Completed	2010	[111]
Ginkgo biloba, alpha-lipoic acid, essential fatty acids, vitamin E, and selenium	MS	n.d.	4 years	II	No results posted	Completed	1999	[112]
Lipoic acid and omega-3 fatty acids	MS with cognitive impairment	54	12 weeks	II	No significant improvement in cognitive function	Completed	2014	[113]
DHA and EPA	Relapsing-remitting MS	50	1 year	IV	Decreased inflammatory markers with no other clinical benefits	Completed	2010	[114]
Melatonin	Relapsing-remitting MS	25	1 year	II	Improved the relapse rate or brain injury; Safe and well tolerated	Completed	2010	[115]
EGCG and coconut oil	MS	60	4 months	II	Decreased inflammatory markers and biomarkers for cardiovascular risk; Decrease in levels of depression	Completed	2018	[116,117,118,119]
Curcumin	Active relapsing MS	80	2 years	II	No neuroprotective effects	Completed	2012	[120]

n.d.—not disclosed.

**Table 4 pharmaceutics-15-00212-t004:** Natural compounds in clinical trials for ALS therapy.

Natural Compound	Condition	Nº of Subjects	Duration	Phase	Main Outcomes	Status	Starting Year	Ref.
Vitamin B7	Probable or confirmed ALS	30	1 year	II	Safe and well tolerated	Completed	2016	[147]
Methylcobalamin (the active form of vitamin B12)	Probable or confirmed ALS	373	3.5 years	II and III	Prolongs survival and retards the progression	Completed	2007	[148]
Vitamin B3	Newly diagnosed or early ALS	300	1 year	n.a.	n.a.	Recruiting	2021	[149]
Vitamin E	Probable or confirmed ALS	32	10 weeks	III	No results posted	Completed	2006	[150]
Vitamin E, NAc cysteine, L-cystine, nicotinamide, and taurursodiol	Probable or confirmed ALS	60	1 year	II	n.a.	Recruiting	2022	[151]
Tretinoin and pioglitazone	Probable or confirmed ALS	28	3 years	I and II	No results posted	Completed	2008	[152]
Theracurcumin	Sporadic or familial ALS	68	6 months	II	n.a.	Completed	2020	[153]
Resveratrol and curcumin in liposomes	Patients with clear diagnosis and symptomatology of ALS since at least 6 months	60	4 months	II	n.a.	Recruiting	2021	[154]
Cannabidiol	Probable or confirmed ALS	25	6 months	IV	n.a.	Not yet recruiting	n.a.	[155]

n.a. not applicable.

**Table 5 pharmaceutics-15-00212-t005:** Natural compounds in clinical trials for HD therapy.

Natural Compound	Condition	Nº of Subjects	Duration	Phase	Main Outcomes	Status	Starting Year	Ref.
Cannabidiol and THC	HD	24	12 weeks	II	No improvement in symptoms	Completed	2011	[163]
EGCG	HD	54	1 year	II	No results posted	Completed	2011	[164]
Resveratrol	HD	102	1 year	n.d.	No results posted	Completed	2015	[165]
Caffeine	HD	100	2 years	n.d.	n.a.	Recruiting	n.a.	[166]
Vitamins B1 and B7	HD with motor symptoms and/or neuropsychiatric	24	52 weeks	II	n.a.	Not yet recruiting	n.a.	[167]

n.a. not applicable; n.d. not disclosed.

## Data Availability

Not applicable.

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
