# Peer review of "Therapeutic Potential of Natural Compounds in Neurodegenerative Diseases: Insights from Clinical Trials"

_pharmaceutics, 2023, doi:10.3390/pharmaceutics15010212_

Round 1

Reviewer 1 Report

Total, this review is important since neurodegenerative diseases (NDs) threaten public health severely and natural products (NPs) are viewed as promising in treating and/or preventing them. The authors systematically introduced the progresses in the studies of natural products used in clinical trial. The paper is well organized and points out the hopes, problems, and some possible resolution in NPs application in NDs. However, some minor revisions are suggested to improve the quality of this paper.

1) In section 2, the authors are suggested to update their references on the overall status of NDs especially about the numbers of NDs patients. For example, the 'world Alzheimer Report 2022' is already online.

2)In the last section, the authors discussed the reasons why some NPs did not show expected significant therapeutic effects such as physiological difference between animals and humans, trials' inherent limitations, and the physical-chemical properties of the compounds. However, as is known, the exact pathological mechanisms of NDs for example AD has not yet explained clearly, which may also lead to the failure of drugs developed on basis of present theories. Thus, some related discussion is also suggested to be supplemented. 

3) There are still some careless mistakes needed to be checked on misspelling, unclear expression, grammar, fonts, and so on.

For example, in Table 1 'down syndrome' should be 'Down syndrome'; 'gringo extract' should be 'Gingko extract'.

On line 111, an 'in' should be inserted before the 'projecting' in 'trouble projecting the voice'.

On line 196, 'However, no differences between the 196 patients were observed' seems to be not consistent with the previous sentences. 

On line 208, the sentence 'Vitamin D3 compound with 208 health-promoting properties obtained mostly through sunlight exposure.' seems to be not correct according to English grammar.

On line 259, 'the compounds ability' should be 'the ability of the compound'.

On line 308, the word 'firmose' should be 'fir moss', 'fir-moss', or 'firmoss'.

Author Response

Total, this review is important since neurodegenerative diseases (NDs) threaten public health severely and natural products (NPs) are viewed as promising in treating and/or preventing them. The authors systematically introduced the progresses in the studies of natural products used in clinical trial. The paper is well organized and points out the hopes, problems, and some possible resolution in NPs application in NDs. However, some minor revisions are suggested to improve the quality of this paper.

Author reply: Thank you for your comments and time spent in dealing with this manuscript. The manuscript was revised accordingly to the comments, and point-by-point answers for each comment are provided below.

1) In section 2, the authors are suggested to update their references on the overall status of NDs especially about the numbers of NDs patients. For example, the 'world Alzheimer Report 2022' is already online.

Author reply: Suggestion accepted. The references have been updated. Please see new references [7], [13], [17], [28], and [29].

2) In the last section, the authors discussed the reasons why some NPs did not show expected significant therapeutic effects such as physiological difference between animals and humans, trials' inherent limitations, and the physical-chemical properties of the compounds. However, as is known, the exact pathological mechanisms of NDs for example AD has not yet explained clearly, which may also lead to the failure of drugs developed on basis of present theories. Thus, some related discussion is also suggested to be supplemented.

Author reply: Suggestion accepted. Please see lines 739-749, page 17.

3) There are still some careless mistakes needed to be checked on misspelling, unclear expression, grammar, fonts, and so on.

Author reply: Suggestion accepted. The article has been thoroughly revised, and corrections are highlighted thought the text.

For example, in Table 1 'down syndrome' should be 'Down syndrome'; 'gringo extract' should be 'Gingko extract'.

Author reply: Suggestion accepted. Please see new Table 1, and line 178 (page 5), lines 301, 302, and 307 (page 7), and line 515 (page 12).

On line 111, an 'in' should be inserted before the 'projecting' in 'trouble projecting the voice'.

Author reply: Suggestion accepted. Please see lines 111, page 3.

On line 196, 'However, no differences between the 196 patients were observed' seems to be not consistent with the previous sentences.

Author reply: Thank you for noticing the mistake. The sentence has been removed.

On line 208, the sentence 'Vitamin D3 compound with 208 health-promoting properties obtained mostly through sunlight exposure.' seems to be not correct according to English grammar.

Author reply: The sentence was corrected. Please see lines 207 and 208, page 5.

On line 259, 'the compounds ability' should be 'the ability of the compound'.

Author reply: Suggestion accepted. Please see line 259, page 6.

On line 308, the word 'firmose' should be 'fir moss', 'fir-moss', or 'firmoss'.

Author reply: Suggestion accepted. Please see line 308, page 7.

Reviewer 2 Report

Andrade and colleagues have performed an extensive and well documented review on the role of natural compounds as possible therapeutic strategies for the treatment of neurodegenerative diseases.

The authors have performed a thorough literature search, and all the clinical trials using natural compounds are highlighted in the study. The tables are clear and the discussion of the studies is well organized and provides a short summary of the best candidates for each diseases.

Author Response

Andrade and colleagues have performed an extensive and well documented review on the role of natural compounds as possible therapeutic strategies for the treatment of neurodegenerative diseases. The authors have performed a thorough literature search, and all the clinical trials using natural compounds are highlighted in the study. The tables are clear and the discussion of the studies is well organized and provides a short summary of the best candidates for each diseases.

Author reply: Thank you for your comments and time spent in dealing with this manuscript.

Reviewer 3 Report

It is an interesting review and worth to be published

Author Response

It is an interesting review and worth to be published.

Author reply: Thank you for your comments and time spent in dealing with this manuscript.